# Rapid Evolution of an Aortic Endocarditis

**DOI:** 10.3390/diagnostics12020327

**Published:** 2022-01-27

**Authors:** Gaetano Todde, Paola Gargiulo, Grazia Canciello, Felice Borrelli, Emanuele Pilato, Giovanni Esposito, Maria Angela Losi

**Affiliations:** Department of Advanced Biomedical Sciences, University Federico II, 80131 Naples, Italy; gaetano.todde@virgilio.it (G.T.); paola.gargiulo@unina.it (P.G.); grazia.canciello@hotmail.it (G.C.); feliceborrelli@yahoo.it (F.B.); emapilato@yahoo.it (E.P.); espogiov@unina.it (G.E.)

**Keywords:** endocarditis, acute aortic regurgitation, transesophageal echocardiography, cardiac surgery

## Abstract

Cardiac surgery is necessary in almost 50% of patients with endocarditis. Early surgery, i.e., the surgery performed during the first hospitalization, is required in the following cases: heart failure secondary to valve regurgitation; *S. aureus*, fungal organism, or other highly resistant organism infection; heart block, annular or aortic abscess, or destructive penetrating lesions; evidence of persistent infection as manifested by persistent bacteremia or fevers lasting >5 days after onset of appropriate antimicrobial therapy. A 62-year-old man developed a fever (38 °C) 3 days after a transaortic electrophysiological study; blood cultures were positive for *S. aureus*, and were sensitive to vancomycin and ceftaroline. Antibiotic therapy was started, controlling the fever and the patient’s infective and inflammatory profiles well; however, 3 days later, acute aortic regurgitation developed. At transesophageal echocardiography (TEE), a rare condition was revealed—vegetation was attached to the aortic wall, impeding correct aortic valve closure. Cardiac operation was carried out and the time for surgery was discussed; based on the patient’s clinically stable condition, and on the infection, which was controlled well by antibiotics therapy, surgery was not performed in emergency circumstance (within 24–48 h)—rather, it was programmed during the hospitalization. A TEE surveillance was initiated, and after 7 days, TEE revealed a new picture, with images of an aortic abscess with small perforation in the right atrium, requiring emergency surgery, carried out 20 h later. In our case, the rapid evolution of the vegetation attached to the aortic wall suggested the following: (1) that the time for the surgery cannot be guided only by clinical procedure but must also be guided by imaging pictures; (2) that strictly TEE surveillance is mandatory in patients with aortic endocarditis not initially referred for emergency surgery.

**Figure 1 diagnostics-12-00327-f001:**
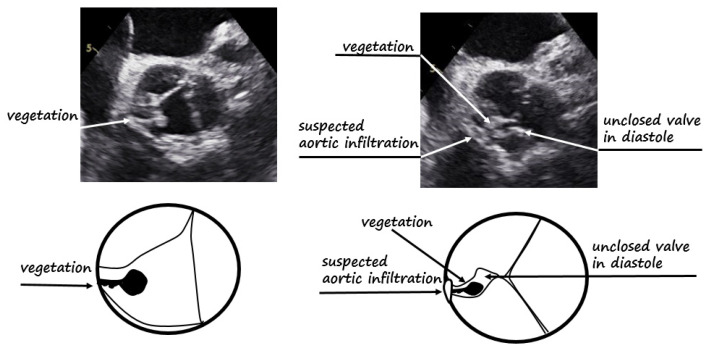
First TEE at the time of acute aortic regurgitation presentation. Top left panel shows TEE at level of aortic valve; a shaggy, pedunculated mass [1] is attached to the aortic wall; bottom left panel shows the relative scheme. Top right panel shows TEE at same level in diastole, with vegetation impeding correct valve closure and a suspected aortic wall infiltration; bottom right panel shows the relative pictogram.

**Figure 2 diagnostics-12-00327-f002:**
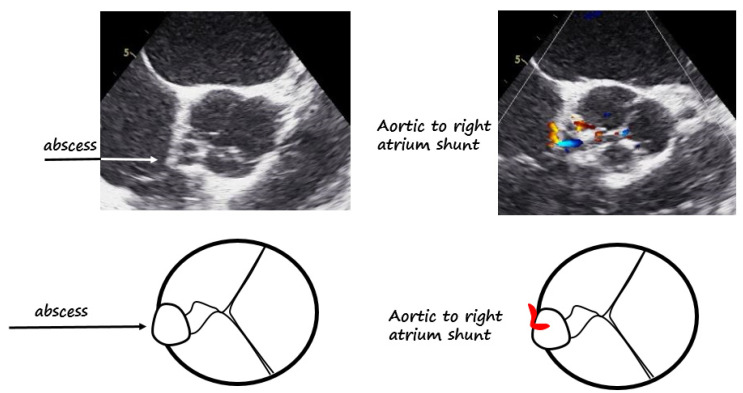
Second TEE, 7 days later. Top left panel shows aortic wall abscess [2]; bottom left panel shows the relative scheme. Top right panel shows TEE at same level with color Doppler, demonstrating a little aortic to the right of the atrium shunt; bottom right panel shows the relative pictogram.

**Figure 3 diagnostics-12-00327-f003:**
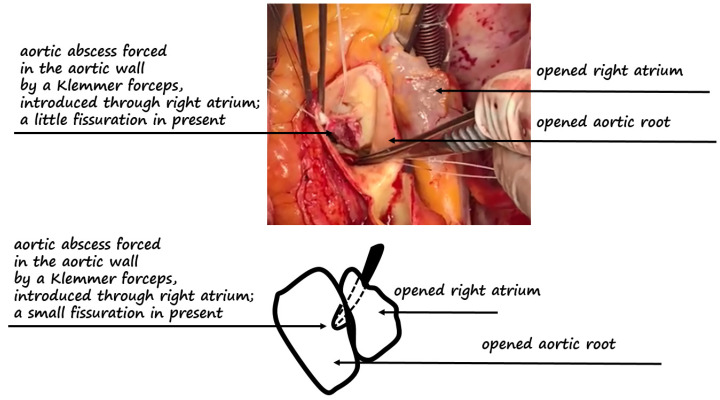
Top panel: surgical view with opened right atrium and aortic root. The Klemmer forceps were forced through the right atrium, showing the aortic abscess with a little fissure [3]. Bottom panel shows the relative pictogram.

## Data Availability

Data related to this study cannot be sent to the outside due to information security policies in the hospital.

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
