# Peer review of "Rapid Evolution of an Aortic Endocarditis"

_diagnostics, 2022, doi:10.3390/diagnostics12020327_

Round 1

Reviewer 1 Report

The authors present a case of aortic endocarditis after endovascular procedure. The case present with septicemia after the procedure and TEE that showed aortic wall vegetation with acute severe AI. At this point, it is not clear why surgery as delayed, even if the patient was stable. The patient would be at risk of systemic embolization with the vegetations attached to the aortic wall. Also, acute severe AI after trans-aortic endovascular procedure and evidence of endocarditis on TEE, would warrant further evaluation for surgery, as AI won't go away with antibiotics.

Repeat TEE then showed invasion of the aortic wall forming an abscess cavity and communication into the right atrium. This complication is a known complication of aortic endocarditis and has been reported previously in the literature

In regard to the conclusion of the report, the patient had TEE initially that showed acute severe AI and endocarditis, so the decision was made with imaging available at the time not only clinical picture. Second, TEE is a well established golden standard tool for evaluation of endocarditis, as stated in the most recent AATS guidelines (reference 3). Thus, this might not be new information to the readers. 

I would suggest that the authors consider resubmit this report as a "case report" with more details about the trans-aortic procedure, could this be iatrogenic injury to the aortic wall that was colonized during bacteremia, what was the details of the decision making to delay surgery after the first TEE, what was the reason to repeat TEE if patient remain stable, what was the organism that cause rapid invasion of the aortic wall … etc.. These details would be helpful to raise the question whether surgery should have not been delayed at the first presentation or not, as a take away message 

Author Response

Reviewer #1.

We appreciate your thoughtful comments and valuable suggestions to improve impact our paper.

We addressed your comments as follows:

  • The authors present a case of aortic endocarditis after endovascular procedure. The case present with septicemia after the procedure and TEE that showed aortic wall vegetation with acute severe AI. At this point, it is not clear why surgery as delayed, even if the patient was stable. The patient would be at risk of systemic embolization with the vegetation attached to the aortic wall.

Also, acute severe AI after trans-aortic endovascular procedure and evidence of endocarditis on TEE, would warrant further evaluation for surgery, as AI won't go away with antibiotics.

We thank the reviewer for his/her comments. We recognized that we did not explain sufficiently medical decision in our patient. In fact, surgery was indicated during hospitalization, becoming emergent when aortic abscess was identified.  We clarified this point reporting ion the presentation of the case the following sentences:

…..Cardiac operation was indicated and time for surgery was discussed: based on clinically stable condition, and on the well-controlled infection by antibiotics therapy, surgery was not per-formed in emergency (within 24-48h), although programmed during the hospitalization…… aortic abscess with small perforation in the right atrium, requiring emergent surgery, carried out 20 hours later……

  • Repeat TEE then showed invasion of the aortic wall forming an abscess cavity and communication into the right atrium. This complication is a known complication of aortic endocarditis and has been reported previously in the literature.

We thank the reviewer for his/her comment. In fact, the corner of our case was not the complication of aortic endocarditis, however what is conditioning the time of surgery in aortic endocarditis.

  • In regard to the conclusion of the report, the patient had TEE initially that showed acute severe AI and endocarditis, so the decision was made with imaging available at the time not only clinical picture.

Probably we did not explain sufficiently our point. Our endocarditis case suggests that, when the aortic wall is involved, even in case of stable clinical condition, emergent surgery could be the right option. So we added a , hopefully, more clear sentence:

 …..In our case, the rapid evolution of the vegetation attached to the aortic wall suggests that: 1) time for surgery cannot be only guided by clinical but must be also guided by imaging pictures;……..  

  • Second, TEE is a well-established golden standard tool for evaluation of endocarditis, as stated in the most recent AATS guidelines (reference 3). Thus, this might not be new information to the readers.

In fact, our case is supporting this recommendation, in that TEE surveillance was performed in our patient.

  • I would suggest that the authors consider resubmit this report as a "case report" with more details about the trans-aortic procedure, could this be iatrogenic injury to the aortic wall that was colonized during bacteremia, what was the details of the decision making to delay surgery after the first TEE, what was the reason to repeat TEE if patient remain stable, what was the organism that cause rapid invasion of the aortic wall … etc.. These details would be helpful to raise the question whether surgery should have not been delayed at the first presentation or not, as a take away message.

We would like to thanks the reviewer particularly for this point: following it, the work in our opinion, is greatly improved.  W enclosed overall information required by the reviewer: …..

….A 62-year-old man, developed fever (38°C) three days after a transaortic electrophysiological study; blood cultures were positive for S. aureus, sensitive to vancomycin and ceftaroline ; anti-biotic therapy was started, well controlling fever and patient’s infective and inflammatory pro-files; however, 3 days later, acute aortic regurgitation developed. At transesophageal echocar-diography (TEE) a rare condition was revealed: vegetation was attached to the aortic wall, im-peding correct aortic valve closure. Cardiac operation was indicated and time for surgery was discussed: based on clinically stable condition, and on the well-controlled infection by antibiot-ics therapy, surgery was not performed in emergency (within 24-48h), although programmed during the hospitalization.  A TEE surveillance was initiated, and after 7 days TEE revealed a new picture with images of aortic abscess with small perforation in the right atrium, requiring emergent surgery, carried out 20 hours later…..

Reviewer 2 Report

The authors report an endocarditis case with special attention to serial (2) transesophageal echocardiographic imaging. The main conclusion is that the decision to cardiac surgery must be made not only clinically but also based on repeated imaging. This is in accordance to the current guidelines.

The TEE images presented are of good quality and the case is interesting though not unique. 

Major Comments-> A) It is not true that surgery is indicated during the first week "in almost all other clinical conditions". Of all endocarditis cases about 50 % undergo cardiac surgery.  B) the imperfect closure of the aortic valve is mostly due to the destruction of valve tissue rather than an incomplete closure because of the vegetations. 

Minor Comments -> The reader would be interested in gender and age of the patient. The type of microorganism that caused the infection is also of utmost interest.

Typos, Spelling -> "(TEE) revealed an interesting a rare condition"; "little perforation" better small perforation; "relative scheme" better pictogram

Author Response

Reviewer #2

We appreciate your thoughtful comments and valuable suggestions to improve impact our paper.

We addressed your comments as follows:

  • The authors report an endocarditis case with special attention to serial (2) transesophageal echocardiographic imaging. The main conclusion is that the decision to cardiac surgery must be made not only clinically but also based on repeated imaging. This is in accordance to the current guidelines. The TEE images presented are of good quality and the case is interesting though not unique. 

We thank the reviewer for his/her consideration concerning our pictures.  We think that vegetation attached to the aortic wall are particularly rare.

2) It is not true that surgery is indicated during the first week "in almost all other clinical conditions". Of all endocarditis cases about 50 % undergo cardiac surgery.  

We thank the reviewer for the comment. We agree that not overall cases of endocarditis undergo surgery. Accordingly, we better clarified what are indication for surgery:

….Cardiac surgery is indicated in almost 50% of patients with endocarditis. Early surgery, i.e the surgery performed during the first hospitalization, is required in case of: heart failure second-ary to valve regurgitation; S. aureus, fungal organism, or other highly resistant organism’s in-fection; heart block, annular or aortic abscess, or destructive penetrating lesions; evidence of persistent infection as manifested by persistent bacteremia or fevers lasting >5 days after onset of appropriate antimicrobial therapy….

3) the imperfect closure of the aortic valve is mostly due to the destruction of valve tissue rather than an incomplete closure because of the vegetation. 

We agree with the reviewer, however, in our case there was a vegetation attached to the aortic wall impeding correct aortic valve closure.

4) The reader would be interested in gender and age of the patient. The type of microorganism that caused the infection is also of utmost interest.

Thanks for these suggestions. We added the information required:

…. A 62-year-old man, developed fever (38°C) three days after a transaortic electrophysiological study; blood cultures were positive for S. aureus, sensitive to vancomycin and ceftaroline ; anti-biotic therapy was started, well controlling fever and patient’s infective and inflammatory pro-files….

5) Typos, Spelling -> "(TEE) revealed an interesting a rare condition"; "little perforation" better small perforation; "relative scheme" better pictogram

Thank you again, we changed the text accordingly.

Other scattered editorial adjustment has also been done.

We hope that our work will meet your expectations.

Round 2

Reviewer 1 Report

The authors have addressed the requested revisions and the value of the report has improved with clear message to the readers